# Structure-guided mutagenesis of OSCAs reveals differential activation to mechanical stimuli

**Sebastian Jojoa-Cruz[1], Adrienne E Dubin[1,2], Wen-Hsin Lee[1], Andrew B Ward[1]\***

[1]Department of Integrative Structural and Computational Biology, Scripps Research, La Jolla, United States; [2]Department of Neuroscience, Scripps Research, La Jolla, United States

**\*For correspondence:**
andrew@scripps.edu

**Competing interest:** The authors declare that no competing interests exist.

**Abstract** The dimeric two-pore OSCA/TMEM63 family has recently been identified as mechanically activated ion channels. Previously, based on the unique features of the structure of OSCA1.2, we postulated the potential involvement of several structural elements in sensing membrane tension (Jojoa-Cruz et al., 2018). Interestingly, while OSCA1, 2, and 3 clades are activated by membrane stretch in cell-attached patches (i.e. they are stretch-activated channels), they differ in their ability to transduce membrane deformation induced by a blunt probe (poking). Here, in an effort to understand the domains contributing to mechanical signal transduction, we used cryo-electron microscopy to solve the structure of *Arabidopsis thaliana* (At) OSCA3.1, which, unlike AtOSCA1.2, only produced stretch- but not poke-activated currents in our initial characterization (Murthy et al., 2018). Mutagenesis and electrophysiological assessment of conserved and divergent putative mechanosensitive features of OSCA1.2 reveal a selective disruption of the macroscopic currents elicited by poking without considerable effects on stretch-activated currents (SAC). Our results support the involvement of the amphipathic helix and lipid-interacting residues in the membrane fenestration in the response to poking. Our findings position these two structural elements as potential sources of functional diversity within the family.

## eLife assessment

The manuscript seeks to dissect the molecular underpinnings of poke and stretch activation in OSCA channels. While the structural and functional experiments are well done, and the authors present some **important** data, the reviewers identified weaknesses in experimental design and interpretation that render the data **incomplete** in supporting some of the main conclusions of the paper. Nevertheless, this work will be of interest to those working in the fields of mechanosensation, sensory biology, and ion channels.

## Introduction

Mechanotransduction allows organisms to gather information about their external surroundings as well as internal processes. Mechanically activated (MA) ion channels are capable of sensing mechanical stimuli and transducing this information to the cell as electrochemical signals (***Ranade et al., 2015***). MA ion channels are diverse and capable of responding to different stimuli such as osmotic stress, membrane deformation, touch, and hearing (***Haswell et al., 2011***; ***Ranade et al., 2015***).

In our previous work, we identified OSCA/TMEM63 as a conserved family of MA ion channels spanning several eukaryotic clades and having considerable diversity in the plant kingdom, where species commonly have more than one paralog. For example, *Arabidopsis thaliana* has 15 members

(*Murthy et al., 2018*) and *Oryza sativa* L. *japonica* (rice), 11 (*Li et al., 2015*). We demonstrated these channels can respond to membrane stretch even more robustly than hypertonicity, originally thought to be the relevant stimulus, with some members (OSCA1.1 and OSCA1.2) also able to respond to poke (*Murthy et al., 2018*). We used cryo-electron microscopy (cryo-EM) to solve the structure of OSCA1.2 in a closed state in both detergent (LMNG) and nanodiscs at high-resolution, identifying a new architecture for MA ion channels that is structurally homologous to TMEM16 proteins (*Jojoa-Cruz et al., 2018*). Results from other groups are consistent with ours (*Liu et al., 2018*; *Maity et al., 2019*; *Zhang et al., 2018*). Similar to the TMEM16 family, OSCAs are dimeric and each subunit has a pore lined by transmembrane helices (TMs) 3–7, instead of a single pore along the symmetry axes (*Jojoa-Cruz et al., 2018*; *Liu et al., 2018*; *Maity et al., 2019*; *Zhang et al., 2018*). A hydrophobic gate on the extracellular half of the pore blocks ion flow. Additionally, molecular dynamics (MD) simulations support our placement of the pore and position lipids along an opening of the pore pathway in the transmembrane domain (TMD; *Jojoa-Cruz et al., 2018*).

Based on the unique structural characteristics of OSCA1.2, we proposed that two features might be involved in sensing membrane deformation: the amphipathic helix (AH) located in the first intracellular loop, and the Beam-Like Domain (BLD), composed of the two cytoplasmic membrane-parallel helices and a hydrophobic hook that connects them and re-inserts itself into the membrane (*Jojoa-Cruz et al., 2018*). Due to the close association of these two elements to the membrane, we reasoned that they could serve as sensors of mechanical stimuli and transmit the signal to the pore through interactions with the TMD. Additionally, a fenestration allows the membrane access to the pore, and MD simulations placed lipids in close association with four positively charged residues in this area suggesting a potential role in gating (*Jojoa-Cruz et al., 2018*). We sought to determine whether these regions contributed to mechanotransduction.

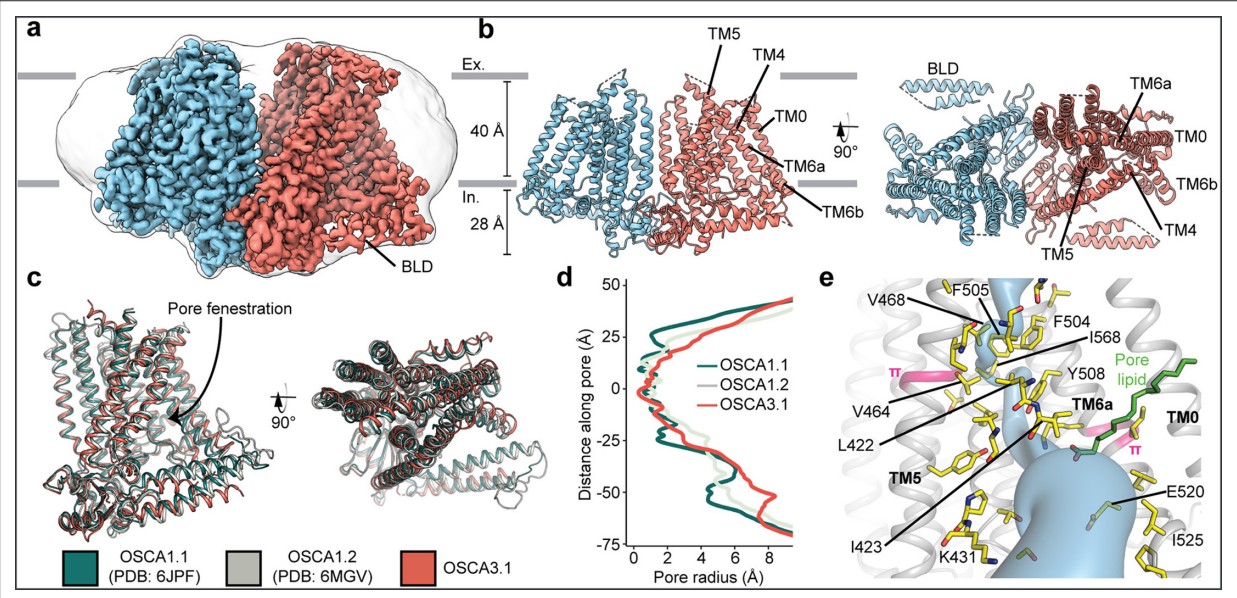

**Figure 1.** Cryo-EM structure and pore of OSCA3.1. (**a**) Cryo-EM map of OSCA3.1 dimer colored by subunit. Nanodisc density in grey corresponds to the unsharpened map (gaussian-filtered to 1.5 σ). (**b**) Front (left) and top (right) view of atomic model. Ex.: Extracellular, In.: Intracellular. (**c**) Superposition of OSCA1.1, OSCA1.2 (in nanodiscs) and OSCA3.1 protomers. (**d**) Pore profile of OSCAs in **c**. (**e**) View of the pore pathway (blue) of OSCA3.1. Pore facing residues colored in yellow, with selected residues labeled. π-helical turns in pink. Putative pore lipid in green. Backbone of TM3 and TM4 helices hidden for clarity.

The online version of this article includes the following figure supplement(s) for figure 1:

**Figure supplement 1.** Purification and cryo-EM data processing of OSCA3.1.

**Figure supplement 2.** Fit of OSCA3.1 model to LocalDeblur map.

**Figure supplement 3.** Comparison of DeepEMhancer and LocalDeblur postprocessed maps.

**Figure supplement 4.** Comparison of OSCA3.1 in nanodiscs and detergent states.

**Table 1.** Data collection, processing, model refinement and validation.

| | OSCA3.1 |
|---|---|
| Data collection and processing | |
| Magnification | 29000 |
| Voltage (kV) | 300 |
| Electron exposure (e–/Å²) | 50 |
| Defocus range (μm) | –0.4 to –1.5 |
| Pixel size (Å) | 1.03 |
| Initial particle images (no.) | 1,913,316 |
| Symmetry imposed | C2 |
| Final particle images (no.) | 197,944 |
| Map resolution (Å) FSC threshold | 2.6 0.143 |
| Map sharpening B factor (Å²) | –64 |
| Model | |
| Composition Peptide chains Protein residues Ligands | 2 1148 24 |
| R.m.s. deviations Bond lengths (Å) Bond angles (°) | 0.023 1.677 |
| Validation MolProbity score Clashscore EMRinger score Poor rotamers (%) | 0.74 0.74 4.12 0.00 |
| Ramachandran plot Favored (%) Allowed (%) Disallowed (%) | 98.93 1.07 0.00 |
| Deposition ID EMDB PDB | 41911 8 U53 |

Here, we used cryo-EM to determine a high-resolution structure of AtOSCA3.1, the first reported OSCA gene (*Kiyosue et al., 1994*), in nanodiscs. In our initial characterization, OSCA3.1 responded to stretch but not poke (*Murthy et al., 2018*). Comparison of the OSCA3.1 structure reported here to previous structures of OSCA1.1 (*Zhang et al., 2018*) and OSCA1.2 (*Jojoa-Cruz et al., 2018*) revealed subtle differences. Structure-function analyses revealed the involvement of the AH and the lipid-interacting residues in the pore pathway of OSCA1.2 in the ability of the channel to respond to membrane deformation by poking. Our results suggest that these features are not only important for activation of OSCA1.2 by poking but may serve to tune the response of OSCA/TMEM63 members to diverse stimuli.

## Results

### Overall architecture of OSCA3.1 is similar to OSCA1.1 and OSCA1.2

We recombinantly tagged OSCA3.1 with EGFP at its C-terminus, expressed the construct in HEK293F cells, and purified it using Lauryl Maltose Neopentyl Glycol (LMNG) supplemented with cholesteryl hemisuccinate (CHS). The EGFP was removed by PreScission Protease cleavage, leaving a stretch of 10 residues on the C-terminus. Purified protein was then reconstituted into nanodiscs and subjected to cryo-EM analysis, resulting in a 2.6 Å resolution symmetric reconstruction (*Figure 1a*, *Figure 1—figure supplement 1*) sufficient to build a model of the majority of the protein (537 out of 724 residues; *Figure 1b–c*, *Figure 1—figure supplement 2*, *Table 1*) Due to increased flexibility of the BLD relative to previous OSCA structures, we were unable to assign residue identity, and instead modeled it as two poly-Ala helices.

Overall, there is good agreement between OSCA3.1, OSCA1.1 and OSCA1.2 protomers (*Jojoa-Cruz et al., 2018*; *Zhang et al., 2018*) (Cα RMSD OSCA3.1-OSCA1.1: 1.243 Å, OSCA3.1-OSCA1.2: 1.152 Å, *Figure 1c*). The pore profile of these OSCAs is maintained and the small radii towards the extracellular side suggests these channels are in a closed/non-conductive state, as suggested by previous MD simulations (*Jojoa-Cruz et al., 2018*; *Zhang et al., 2018*; *Figure 1d*). Likewise, the two π-helical turns in TM5 and TM6a seen in previous structures near the neck of the pore are present at similar positions in OSCA3.1 (*Jojoa-Cruz et al., 2018*; *Maity et al., 2019*). We identified multiple

lipid-like densities in one of our postprocessed maps (*Figure 1e*, *Figure 1—figure supplement 3*). Notably, one of these densities is located at the pore fenestration, at a position similar to that observed in previous MD simulations of OSCA1.2 (*Jojoa-Cruz et al., 2018*), supporting the proposed occupation of the pore pathway by lipids. Recent cryo-EM structures of AtOSCA1.2 (*Jojoa-Cruz et al., 2024*), and AtOSCA1.1 and human TMEM63A (*Zhang et al., 2023*) also place a lipid at a similar position.

Previously, the structure of OSCA3.1 was solved in a digitonin micelle (*Zhang et al., 2018*), and more recently, in LMNG (*Zhang et al., 2023*). The similarity of our nanodisc sample to these detergent structures suggests that the use of detergent did not affect the conformation or oligomerization of OSCA3.1, similar to previous results with OSCA1.2 (*Jojoa-Cruz et al., 2018*; *Liu et al., 2018*; *Figure 1—figure supplement 4a–b*). However, the inter-subunit cleft in the nanodisc structure is wider due to outward movements of innermost TMs ranging from ~4–8 Å (*Figure 1—figure supplement 4c*) relative to the digitonin sample, which was recently denoted as an 'extended' state (*Zhang et al., 2023*). Thus, our nanodisc structure represents a further 'extended' state than the one found both in digitonin and LMNG micelles (*Zhang et al., 2023*; *Figure 1—figure supplement 4d*). Whether these differences represent biologically meaningful states, an effect of the lipid environment (detergent vs lipids), or inherent flexibility of the channel remains to be determined. Additionally, our structure enabled a higher degree of certainty in the residue assignment relative to the initial digitonin structure, particularly for TM0 where sidechains were not previously modelled (*Zhang et al., 2018*). Consequently, we have shifted the registry of TM0 by two residues relative to the detergent structure, in agreement with the LMNG structures (*Zhang et al., 2023*).

## Mutation of key residues in the amphipathic helix abrogates poke but not stretch responses in OSCA1.2

Structures of AtOSCA1.1, rice and At OSCA1.2, and OSCA3.1 (*Figure 2a*, *Figure 2—figure supplement 1*), as well as sequence alignment, suggest that the AH between TM0 and TM1 of OSCAs is a conserved feature of these channels. AHs can serve as both membrane anchors and sensors of membrane deformation (*Drin and Antonny, 2010*), and they are present in many MA ion channels (*Bavi et al., 2016*; *Brohawn et al., 2014a*; *Jojoa-Cruz et al., 2018*; *Saotome et al., 2018*). There is evidence for their involvement in mechanical force transduction; for example, in the bacterial mechanosensitive channel of large conductance MscL, an AH in the cytoplasmic side of the bilayer links membrane tension to protein conformation (*Figure 2—figure supplement 2a and b*; *Bavi et al., 2016*). Interestingly, structures of members of the TMEM16 family, structural homologs of OSCAs, have a similarly placed helix in this region; however, it is located in the cytosol and does not present amphipathic properties (*Figure 2—figure supplement 1c-e*; *Alvadia et al., 2019*; *Bushell et al., 2019*; *Paulino et al., 2017*). Involvement of TMEM16A in the detection of mechanical forces in myocytes and bile ducts was dependent on calcium influx (potentially caused by an upstream MA cation channel) and not a direct response to the mechanical stimulus (*Bulley et al., 2012*; *Dutta et al., 2013*). To our knowledge, characterization of TMEM16 proteins as MA ion channels has not been reported. Based on these observations, we mutated the membrane-facing residues at each end of the AH of OSCA1.2 in an effort to decrease its propensity to interact with the membrane. Single (W75K or L80E) and double (W75K/L80E) mutants in OSCA1.2 strongly abrogated the response to indentation of the membrane with a blunt glass probe (poke) (WT: 47 of 51 cells produced MA currents; W75K: 3 of 24; L80E: 2 of 16; W75K/L80E: 2 of 24; *Figure 2b*, top panel). Moreover, in OSCA1.2$_{W75K/L80E}$, a larger displacement of the membrane by the probe relative to the WT appeared to be needed to observe currents ('apparent threshold'; WT: 5.3±0.5 µm (N=46; mean ± S.E.M.), W75K/L80E: 15.3±2.3 µm (N=2), with the none-responsive W75K/L80E cells rupturing at a 13.0±0.5 µm (N=22); *Figure 2b*, bottom panel). The insensitivity to the poking stimulus was not due to poor membrane trafficking of these constructs (see below). The loss of sensitivity of cells expressing these mutant constructs to poking suggests that the AH of OSCA1.2 plays an important role in this mechanically activated response.

We then tested whether these mutations in the AH affected the response of OSCA1.2 to stretch. Contrary to what was observed during poking, all cell-attached patches revealed stretch-activated currents (WT: 25 of 26 cells; W75K: 3 of 3; L80E: 5 of 5; W75K/L80E: 11 of 11; *Figure 2c*, top panel; electrode resistances were similar in all cases [see Figure legend]). Despite differences in the amplitude of macroscopic currents, which could be explained by variation in expression levels, all mutant channels were activated at similar thresholds (WT: –48±5 mmHg (N=25); W75K: –56±11 mmHg

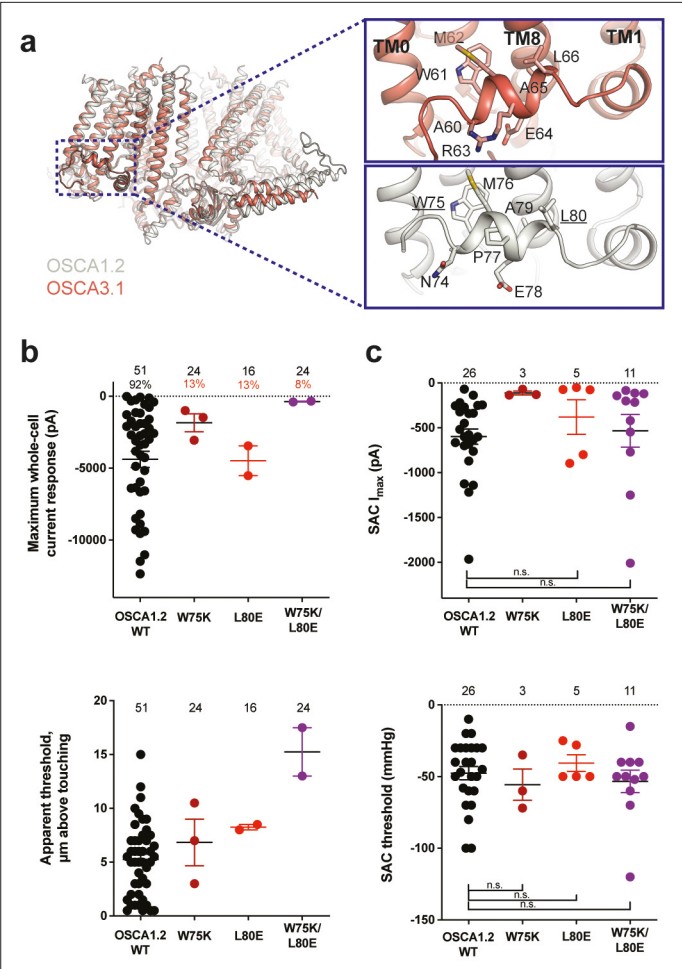

**Figure 2.** Amphipathic helix mutants of OSCA1.2. (**a**) Superposition of OSCA1.2 (grey) and OSCA3.1 (red). Insets: close-up view of amphipathic helix. Residues substituted in OSCA1.2 for electrophysiology experiments are underlined. (**b**) Few cells expressing mutations in the amphipathic helix respond to poking compared to WT controls. Top panel: Maximum poke-induced currents observed in whole-cell mode for cells exposed to displacements up to 8.9±0.6 µm (N=51; mean ± S.E.M.), 12.8±0.7 µm (N=24), 12.4±0.7 µm (N=16), and 12.8±0.5 µm (N=24) above touching for WT, OSCA1.2$_{W75K}$, OSCA1.2$_{L80E}$, and OSCA1.2$_{W75K/L80E}$, respectively. Bottom panel: the apparent threshold in µm above touching the cell for this cohort. (**c**) SAC maximal current ($I_{max}$) (top) and mmHg threshold (bottom) from WT and OSCA1.2$_{W75K/L80E}$-expressing cells reveal no significant differences in the ability of negative pressure to activate channels in cell-attached patches (Student's *t*-test). Also shown are data from single mutants OSCA1.2$_{W75K}$ and OSCA1.2$_{L80E}$. Too few patches were obtained for OSCA1.2$_{W75K}$ to compare $I_{max}$. Electrode resistances for (**c**) were similar in all cases (WT: 2.3±0.1 MΩ (N=26); W75K: 2.1±0.4 MΩ (N=3), L80E: 2.2±0.1 MΩ (N=5), W75K/L80E: 1.8±0.1 MΩ (N=11)). For panels (**b–c**): individual cells are represented as scatter points; mean and S.E.M. are displayed.

The online version of this article includes the following figure supplement(s) for figure 2:

**Figure supplement 1.** Protein sequence alignment of OSCA3.1.

**Figure supplement 2.** Comparison of the amphipathic helix of OSCA3.1 with corresponding helices in TMEM16 structures.

**Figure supplement 3.** Poke-induced response of OSCA1.2$_{P77R}$.

**Figure supplement 4.** OSCA3.1 may be poke-sensitive at high thresholds.

(N=3), L80E: –41±6 mmHg (N=5), W75K/L80E: –53±8 mmHg (N=11)) (*Figure 2c*, bottom panel). In all cases, there was variability in the rate of inactivation from moderately fast to extremely slow (data not shown). Overall, the observed stretch response of OSCA1.2 was apparently not affected by the mutations we introduced.

Subsequently, we mutated the proline in the OSCA1.2 AH to the corresponding arginine in OSCA3.1 to determine whether this difference impacts the ability of OSCA3.1 to robustly transduce the poke stimulus; this change (OSCA1.2$_{P77R}$) had no effect on the poke response (*Figure 2—figure supplement 3*).

It is worth noting that in our initial characterization, OSCA3.1-expressing cells produced SAC while poke did not induce observable currents (*Murthy et al., 2018*). However, in the present experiments we were able to record poke-activated currents mediated by OSCA3.1 on a few occasions (OSCA1.2$_{WT}$: 6 of 6 cells; OSCA3.1$_{WT}$: 3 of 10) (*Figure 2—figure supplement 4a*). The apparent threshold for OSCA3.1 activation by poke-induced membrane displacement tended to be higher than for OSCA1.2 but the difference was not statistically significant (OSCA1.2$_{WT}$: 10.4±1.2 μm (N=6; mean ± S.E.M.); OSCA3.1$_{WT}$: 13.8±0.9 μm (N=3)) (*Figure 2—figure supplement 4b*), perhaps due to the low numbers of responsive cells analyzed. These data suggest that the poking stimulus is capable of activating OSCA3.1 but the stimulus intensity required is close to the rupture point of the HEK293T-Piezo1-knockout (HEK-P1KO) cells used in our assay.

## Replacing the OSCA3.1 Beam-Like Domain (BLD) in OSCA1.2 had no effect on stretch-induced currents but decreased the apparent sensitivity of the cells to poke

One of the unique features of OSCAs is the presence of the BLD, a hydrophobic hook that, by inserting itself into the membrane, may serve as an anchor that could potentially be displaced under tension (*Jojoa-Cruz et al., 2018*; *Liu et al., 2018*; *Maity et al., 2019*). Hydrogen/deuterium exchange mass spectrometry (HDXMS) of rice OSCA1.2 indicated that the helix of the BLD closest to the membrane undergoes less deuterium exchange and potentially remains associated to a neighboring surface (*Maity et al., 2019*), hinting at an important interaction with TM6b. In OSCA3.1, the BLD is poorly resolved, likely due to flexibility (*Figure 1c*). To test whether the increased flexibility in this region affects MA responses, we replaced the BLD of OSCA1.2 with the BLD of OSCA3.1 (*Figure 3a*). HEK-P1KO cells expressing the chimera OSCA1.2$_{OSCA3.1-BLD}$ produced similar MA currents in both poke and stretch assays (*Figure 3b and c*) with the exception that a slightly stronger displacement stimulus was required to activate the mutant channels (*Figure 3b*, bottom panel). These data suggest that the BLD is not responsible on its own for the ability of the channels to be activated by poking, however, it appears to play a role. As expected for similar SAC from cells expressing OSCA1.2 and OSCA3.1, there was no obvious change in SAC when the OSCA1.2 BLD was replaced with the OSCA3.1 BLD. It should be noted that the BLD tolerates a certain degree of mutation as sequence alignment in this region shows higher variability than other secondary structures of OSCAs (*Figure 2—figure supplement 1*). Thus, we cannot rule out that in the OSCA1.2 background, the BLD of either channel is capable of performing a similar role.

## Substitution of potential lipid-interacting lysine residues for isoleucine abrogates the poke response in OSCA1.2

Lipids play an important role in the stability and function of ion channels (*Duncan et al., 2020*). In TRAAK, a eukaryotic mechanosensitive channel, occlusion of the pore by a lipid acyl chain has been proposed as a gating mechanism (*Brohawn et al., 2014a*; *Brohawn et al., 2014b*). In the volume-regulated channel SWELL1 lipids block the pore in the closed state (*Kern et al., 2023*). In the bacterial mechanosensitive channel MscS, the occupation of lipid pockets may determine channel conformation (*Pliotas et al., 2015*; *Zhang et al., 2021*), and in FLYC1, a Venus flytrap homolog of MscS, conformational changes may allow lipids to access the pore and occlude ion conduction (*Jojoa-Cruz et al., 2022*). Moreover, exposure of inside-out patches from OSCA1.1-expressing HEK cells to lyso-phosphatyidylcholine (LPC) enhanced channel response to stretch (*Zhang et al., 2018*).

Previously, we identified four lysine residues in OSCA1.2 located in TM4 and TM6b at the pore fenestration. These residues interacted with lipid phosphate head groups throughout the duration of our MD simulations (*Jojoa-Cruz et al., 2018*). The OSCA3.1 structure revealed that two of these lysine residues (K435 and K536 of OSCA1.2) have been replaced by isoleucine (I423 and I525, respectively), which are expected to reduce the interaction with lipid phosphate head groups (*Figure 4a*). To test for the contribution of these lipid-interacting residues to channel mechanosensitivity, we measured poke- and stretch-induced currents in the double mutant OSCA1.2$_{K435I/K536I}$ (*Figure 4b and c*). The double

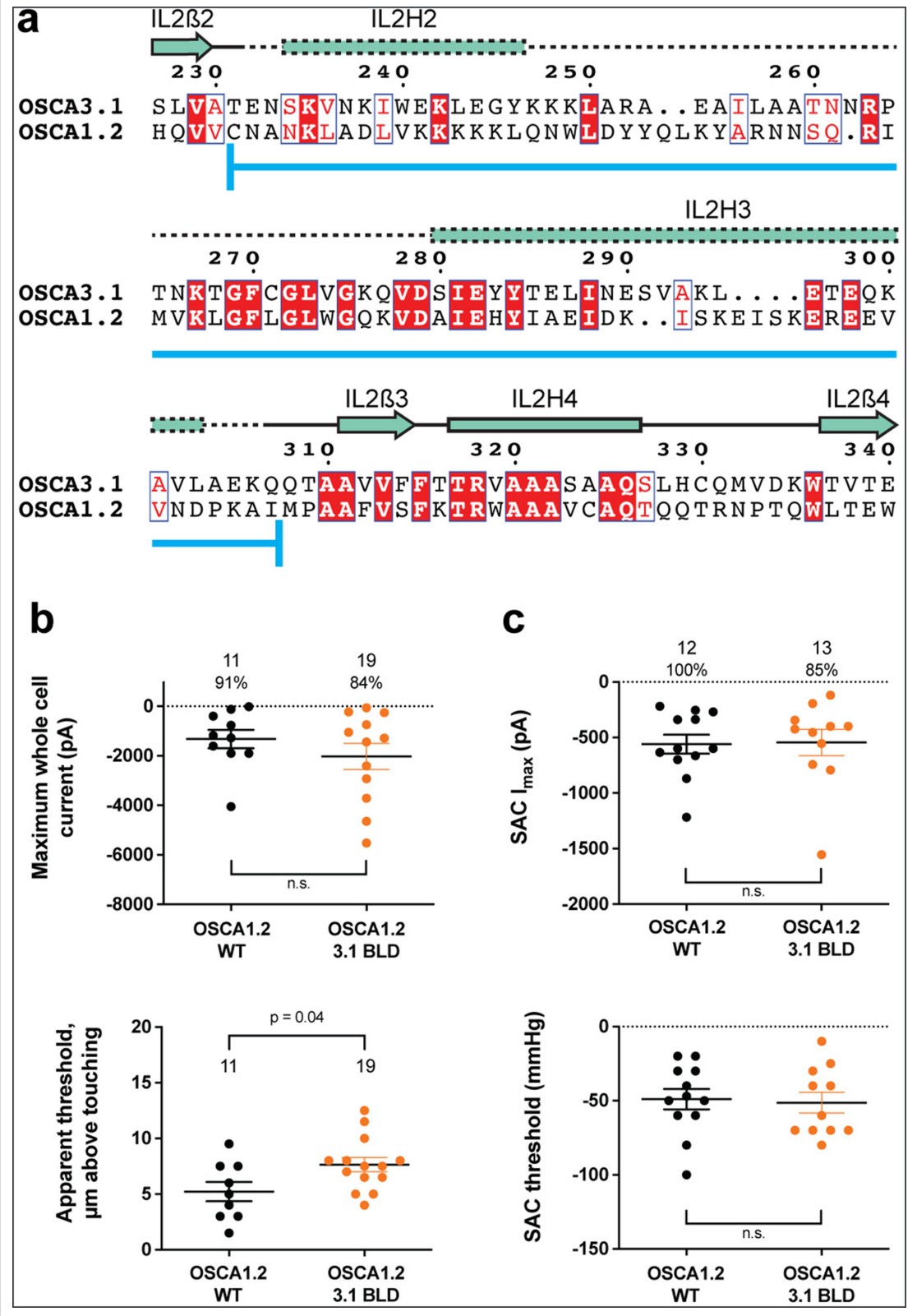

**Figure 3.** OSCA1.2$_{OSCA3.1-BLD}$ chimera. (**a**) Amino acid sequence alignment of the BLD region of OSCA1.2 and OSCA3.1. Full alignment in *Figure 2—figure supplement 1*. Blue line at bottom of sequences denotes the sequence swapped in OSCA1.2$_{OSCA3.1-BLD}$ chimera. (**b**) Poke-induced responses are observed in HEK-P1KO cells expressing OSCA1.2$_{3.1-BLD}$. Top panel: Maximum poke-induced currents observed in whole-cell mode for cells exposed to displacements up to 9.8±0.8 µm (N=11; mean ± S.E.M.) and 11.8±0.6 µm (N=19) above touching for WT and OSCA1.2$_{3.1-BLD}$, respectively. The

*Figure 3 continued on next page*

*Figure 3 continued*
percentage of cells with responses are shown above. Bottom panel: the apparent threshold in μm above touching the cell for this cohort. (**c**) SAC I<sub>max</sub> (top) and mmHg threshold (bottom) from WT and OSCA1.2<sub>3.1BLD</sub>-expressing cells reveal similar activity induced by negative pressure in cell-attached patches. Data shown were obtained from the same experiments. Electrode resistances for (**c**) were similar in all cases (WT: 2.4±0.1 MΩ (N=12); OSCA1.2<sub>OSCA3.1-BLD</sub>: 2.5±0.1 MΩ (N=13)).

mutant responded poorly to poking, with only 1 out of 20 cells producing currents upon mechanical stimulation (K435I/K536I: –365 pA (N=1); *Figure 4b*, comparison made to WT data acquired on the same days). This finding was not due to poor expression of the double mutant in HEK-P1KO cells; the SAC I$_{max}$ observed in cell-attached patches was similar to WT (*Figure 4c*, top panel). Furthermore, the mutant could be activated by stretch at similar thresholds to WT (WT: –43±10 mmHg (N=6); K435I/K536I: –60±10 mmHg (N=7); *Figure 4c*, bottom panel), and similar stretch-response dependence (i.e. similar pressures required to elicit half-maximal currents (P$_{50}$) and slope; *Figure 4d*). Although macrocurrent amplitudes in SAC recordings tended to be lower than WT tested on the same day, the ~2 fold difference unlikely accounts for the ~20-fold fewer responsive cells (*Figure 4b and c*). Taken together, these results suggest that loss of positively charged sidechains in the fenestration selectively impairs poke-induced responses of OSCA1.2 under physiological conditions, but not its responsiveness to the stretch stimulus.

## Discussion

Here, we performed structure-function studies to determine the molecular underpinnings of mechanical activation of OSCA ion channels. We solved the structure of OSCA3.1 at a resolution of 2.6 Å in a non-conductive state and revealed a marked structural conservation to previous OSCA orthologs. The

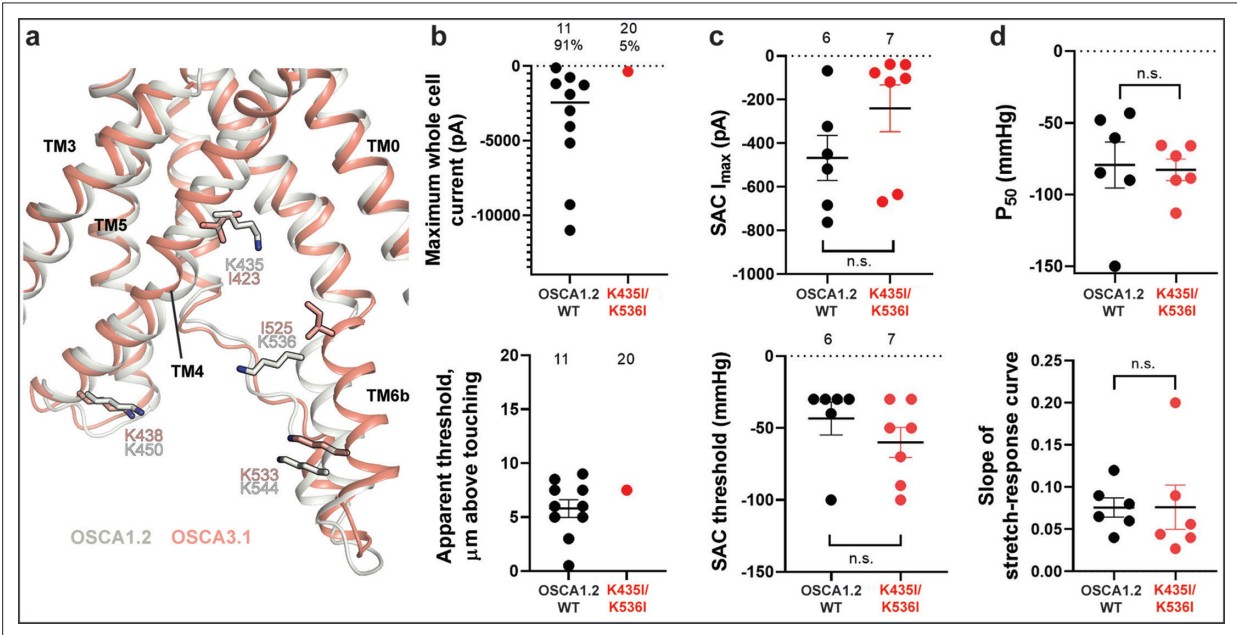

**Figure 4.** Probing the functional role of the potential lipid-interacting residues. (**a**) Superposition of OSCA1.2 and OSCA3.1 around the pore fenestration. Residues of OSCA1.2 predicted to interact with lipids and corresponding residues in OSCA3.1 are shown. (**b**) Few OSCA1.2$_{K435I/K536I}$-expressing HEK-P1KO cells respond to the poke stimulus while nearly all cells expressing OSCA1.2 WT channels respond (percentages shown above under the N of cells tested). Top panel: Maximum poke-induced currents observed in whole-cell mode for cells exposed to displacements up to 9.7±1 μm (N=11; mean ± S.E.M.) and 12.1±0.6 μm (N=20) above touching for WT and OSCA1.2$_{K435I/K536I}$, respectively. Bottom panel: the apparent threshold in μm above touching the cell for this cohort. (**c**) SAC Imax (top) and mmHg threshold (bottom) from WT and OSCA1.2$_{K435I/K536I}$-expressing cells reveal no significant differences in the ability of negative pressure to activate channels in cell-attached patches (Student's *t*-test). Electrode resistances were similar in all cases (WT: 2.1±0.2 MΩ (N=10); K435I/K546I: 1.7±0.3 MΩ (N=7)). Data were collected within 15 min of exposure of cells to high K+ used in this assay. (**d**) Analysis of stimulus-response relationships reveal no significant differences (Student's *t*-test) in pressure to half-maximal activation (P$_{50}$; top) or the slope of the curve (bottom). Whole cell and SAC data were obtained from cells transfected at the same time.

final map presented several lipid-like densities surrounding the channel, including one density close to the pore and several densities populating the inter-subunit cleft. This ion channel was reported to be activated by stretch in cell-attached patches but insensitive to a poke stimulus using a blunt glass probe (*Murthy et al., 2018*). Comparisons between the structures of OSCA1.2, which is robustly activated by both poke and stretch, to OSCA3.1, which is only robustly activated by stretch, suggested three domains that might contribute to gating during a poke stimulus. First, a conserved short amphipathic helix (AH) which sits at the membrane interface reminiscent of a similar feature in other MA ion channels like MscL (*Bavi et al., 2016*), Piezo1 (*Saotome et al., 2018*), MscS (*Reddy et al., 2019*), and TRAAK (*Brohawn et al., 2012*). Second, side chains in TM4 and TM6b in the vicinity of the membrane fenestration may have differential interactions with lipids as indicated by MD simulations of OSCA1.2. Third, the beam-like domain (BLD) that contains two helices and linker that hooks around the cytoplasmic surface into the membrane in OSCA1.2 but is less well-resolved in the OSCA3.1 map presumably because it is not intimately associated with the channel/membrane. Mutagenesis of the AH and residues at the fenestration in OSCA1.2 largely abrogated its response to poking without affecting its activation by stretch; replacing the BLD with that from OSCA3.1 only slightly impaired the response to poking, having no effect on stretch-induced activation.

Residues in the AH were mutated to either confer more of an OSCA3.1 character or tend to make the domain less likely to interact with the membrane, similar to the TMEM16 family that are not thought to be directly activated by mechanical stimuli. Swapping the divergent residue in the AH of OSCA1.2 with its equivalent in OSCA3.1 (OSCA1.2$_{P77R}$), had no negative impact on the sensitivity to poking. These data suggest that the difference in poke sensitivity between OSCA1.2 and OSCA3.1 is not due to the AH itself. However, when mutations were incorporated to presumably destabilize the interaction of the AH domain with the membrane (OSCA1.2$_{W75K/L80E}$), fewer cells could be activated by poking to similar stimulus strengths while stretch-activation of channels was not impaired. Although a WT OSCA1.2 AH is critical for normal poke-induced channel opening, these mutations do not hinder gating by membrane stretch.

Substituting the BLD of OSCA1.2 for that of OSCA3.1 had little effect on poke- or stretch-activated responses. Although these results suggest that the BLD may not be involved in modulating the MA response of OSCA1.2, a recent report suggested evolutionary coupling of three sets of residues in the BLD and TM6b that may form electrostatic interactions between these domains (*Jojoa-Cruz et al., 2024*). The evolutionarily favored residue identities at these sites constitute three potential salt bridges: **R/K**257-**E**557, **E**260-**R**553, and **R/K**261-**E/D**554 (based on OSCA1.2 numbering). These sites in OSCA1.2$_{OSCA3.1-BLD}$ chimera correspond to N257**K-E**557, D260**A-K**553, and Y261**R-D**554 (mutation-interacting residue), perhaps leaving intact two out of three potential salt bridges and maintaining its influence (*Figure 2—figure supplement 1*). It is worth noting that a similar chimera of AtOSCA1.2 with the IL2 domain of the human ortholog TMEM63A (OSCA1.2$_{63A\ IL2}$), containing most of the intracellular domain which includes the BLD as well as the dimerization domain, presented little inactivation of SAC akin to monomeric TMEM63A$_{WT}$ (*Zheng et al., 2023*). This behavior was not due to an inability of OSCA1.2$_{63A\ IL2}$ to dimerize, as the monomeric OSCA1.2$_{5Mu}$, obtained by disrupting the dimer interface with five mutations while maintaining OSCA1.2$_{WT}$ BLD, exhibited pressure-induced SAC gating similar to OSCA1.2$_{WT}$ (*Zheng et al., 2023*). This new evidence, taken together with the decrease in the apparent poking sensitivity of the OSCA1.2$_{OSCA3.1-BLD}$ chimera in this study, suggests that the BLD plays a role in the force-induced gating of the OSCA family.

The structure of OSCA3.1 exhibits an open fenestration similar to OSCA1.2 and lipids are expected to populate this area (*Jojoa-Cruz et al., 2018*), similar to lipid occlusion of the pore in other MA ion channels such as TRAAK (*Brohawn et al., 2014a*) and MscS (*Pliotas et al., 2015*; *Zhang et al., 2021*). Our results suggest that TM4 and TM6b interactions with lipids may contribute to activation during poking. Modification of OSCA1.2 residues to match those of OSCA3.1 (OSCA1.2$_{K435I/K536I}$) produced an OSCA3.1-like phenotype (MA currents are observed in response to stretch but rarely to poke). These mutant channels are not completely insensitive to poking but appear to have a very high threshold to activate, one that is often beyond the point of cell rupture by the stimulus (similar to that seen with OSCA1.2$_{W75K/L80E}$). Notably, TM4 in TMEM16 lipid scramblases and TM6 in both TMEM16 scramblases and channels undergo conformational changes during activation (*Alvadia et al., 2019*; *Kalienkova et al., 2019*; *Paulino et al., 2017*). The loss of positive charges in OSCA3.1 may weaken the interactions between residues and lipids, similar to MscS where charge neutralization of the lipid facing

residue R59 by the introduction of a leucine strongly altered gating, potentially by disturbing a salt bridge with a lipid phosphate (*Flegler et al., 2021*; *Rasmussen et al., 2019*). A weaker interaction could decrease the tension generated by the lipids upon mechanical stimulus, hence increasing the force needed to open the channel, as seen in our poke experiments. A similar argument could be made for the AH of OSCAs: by disturbing its coupling with the lipid bilayer, a greater stimulus might be needed to activate the channel. It is also important to note that the membrane of a plant cell contains a different lipid composition than that of HEK293 cells used in our assays, and thus these lipids, or association with the plant cell wall, may alter how these channels respond to physiological stimuli in vivo.

The phenotypes of OSCA1.2$_{W75K/L80E}$ and OSCA1.2$_{K435I/K536I}$ suggest a mechanism recruited by poking is no longer able to adequately activate OSCA1.2$_{W75K/L80E}$ and OSCA1.2$_{K435I/K536I}$ channels. Since the factors contributing to channel activation in the poking assay are not understood, we can only speculate as to mechanisms underlying the loss of sensitivity to poking in these mutants. Our results suggest that these stimuli are sensed by different features of OSCA channels, akin to the well-studied ThermoTRP channels (*Bandell et al., 2006*; *Fernández-Ballester et al., 2023*; *Jordt and Julius, 2002*). Polymodal activation mechanisms have been recently reported for the MA channel PIEZO2 where a mutagenesis approach revealed differential effects on the ability to respond to poking and stretch (*Verkest et al., 2022*). Nonetheless, the discrepancy could be due to inherent methodological differences between these two assays, as whole-cell recordings during poking involve channels in inaccessible membranes (at the cell-substrate interface) and channel interactions with extracellular and intracellular components (*Richardson et al., 2022*), while the stretch assay is limited to recording channels inside the patch.

Overall, the analysis of these mutants demonstrates the involvement of the AH and residues at the membrane fenestration in the MA response of OSCAs, as we initially proposed (*Jojoa-Cruz et al., 2018*). The participation of residues from TM4 and TM6 hint at further similarities with TMEM16 family, but to what extent is unknown. Perhaps the most surprising result is the selective disruption of poke-activated currents of OSCA1.2 without affecting its response to membrane stretch, which suggests that changes in these features could serve to fine-tune the response to certain mechanical stimuli and may be a source of functional diversity within this large family. The structures and characterizations presented here and in our previous work (*Jojoa-Cruz et al., 2018*; *Murthy et al., 2018*) will prove fundamental in assessing the physiological consequences of these differences and further understanding the role of the OSCA/TMEM63 family.

## Materials and methods

### Expression constructs

For structural studies, the OSCA3.1 (UniProt ID: Q9C8G5) coding sequence was synthesized using optimized codons for expression in human cells and subsequently cloned into the pcDNA3.1 vector used in our previous report (*Jojoa-Cruz et al., 2018*). In this plasmid, the C terminus of the protein sequence is followed by a PreScission Protease cleavage site, EGFP and a FLAG tag.

### Protein expression and purification

Samples of OSCA3.1 in nanodisc were obtained following the same method reported previously for OSCA1.2 in nanodiscs (*Jojoa-Cruz et al., 2018*), with the exception of the detergent used. Instead of beta-D-maltopyranoside (DDM), we used Lauryl Maltose Neopentyl Glycol (LMNG) due to increased yields. Thus, the detergent composition of the solubilization and wash buffers was 1%/0.1% and 0.01%/0.001% LMNG/cholesteryl hemisuccinate (CHS), respectively. Fractions from Size Exclusion Chromatography (SEC) corresponding to the OSCA3.1 in nanodiscs peak were concentrated to 3.2 mg/mL (corrected using the extinction coefficient of OSCA3.1).

### Cryo-EM sample preparation and data collection

A total of 3.5 µL of OSCA3.1 at 3.2 mg/mL were frozen using the same conditions as OSCA1.2 samples (*Jojoa-Cruz et al., 2018*). Grids were imaged on a Titan Krios (Thermo Fisher) operating at 300 kV with a K2 Summit direct electron detector (Gatan), with pixel size of 1.03 Å (nominal magnification of 29,000 x). A total of 38 frames were collected per movie, adding to a total dose of ~50 electrons per

Å². Automated image collection was done through Leginon (*Suloway et al., 2005*) using a defocus range of –0.4 to –1.5 µm. A total of 8375 movies were collected.

## Cryo-EM image processing

During collection, movies were aligned and dose-weighted using MotionCor2 (*Zheng et al., 2017*). The resulting micrographs were imported into cryoSPARCv2.5 (*Punjani et al., 2017*) and CTF values were estimated through Gctf (*Zhang, 2016*). Approximately 300 particles were manually picked from ~10 micrographs and subjected to 2D classification to obtain templates. Template picking was tested on 20 micrographs and particles used to generate better 2D templates. These templates were later used to pick particles on ~1600 micrographs. The resulting ~900 k particles were subjected to, in order, 2D classification, ab initio reconstruction, two rounds of heterogeneous refinement, homogeneous refinement and non-uniform refinement to obtain an initial 3D reference, which reached high-resolution. Unless otherwise specified, C2 symmetry was applied to all homogeneous, non-uniform and RELION 3D refinements.

For the full dataset, 8375 movies were imported into cryoSPARCv2 and subjected to full-frame motion correction followed by Gctf estimation. Given the large size of the dataset, and the fact that initial processing had returned a high-resolution structure, we decided to apply a 2.6 Å CTF cutoff to reduce the number of micrographs to 3068 and speed up processing. Templates were generated based on the initial 3D reference. Template picking, followed by local motion correction and 2D classification resulted in 1,913,316 selected particles for our initial stack. Ab initio reconstruction produced three classes, and heterogeneous refinement was performed on the two worst classes. The resulting best class was combined with the best class from ab initio and used for two rounds of heterogeneous and homogeneous refinements. The heterogeneous refinement of the first round was done without symmetry, from that point onwards, C2 symmetry was imposed unless otherwise specified. Only particles belonging to the best class were selected for further processing (318,249). Non-uniform refinement, followed by 2 rounds of local refinement (without imposing symmetry), were performed on these particles before exporting them into RELION-3.1 (*Scheres, 2012*; *Zivanov et al., 2020*). The particle stack underwent the following process in RELION. 3D refinement, CTF refinement, 3D refinement with global angular searches, 3D refinement using SIDESPLITTER (SS) (*Ramlaul et al., 2020*), and two rounds of CTF and 3D refinements using SS. Unless specified, all 3D refinements were limited to local angular searches. 3D classification without alignment (no imposed symmetry) was performed on the particle stack and the best class, comprising 197,944 particles, was subjected to a final 3D refinement with SS. The map was sharpened separately with LocalDeblur (*Ramírez-Aportela et al., 2020*) through Scipion (*de la Rosa-Trevín et al., 2013*; *de la Rosa-Trevín et al., 2016*; *Vilas et al., 2018*) and DeepEMhancer using the 'highRes' model (*Sanchez-Garcia et al., 2021*). The FSC and local resolution of the map was calculated through RELION.

## Model building and refinement

A homology model for OSCA3.1 was obtained through SWISS-MODEL (*Waterhouse et al., 2018*), using as template the structure of OSCA1.2 in nanodiscs (PDB: 6MGV) (*Jojoa-Cruz et al., 2018*). Iterative rounds of building in Coot (*Casañal et al., 2020*; *Emsley and Cowtan, 2004*) and real space refinement in Phenix (*Afonine et al., 2018b*; *Adams et al., 2019*) and Rosetta (*Wang et al., 2016*) were used to generate the final model. SMILES codes for the ligands were used to obtain the appropriate restrains using eLBOW (*Moriarty et al., 2009*). Model validation was carried out with MolProbity (*Williams et al., 2018*) and EMringer (*Barad et al., 2015*) to the LocalDeblur map. Phenix mtriage (*Afonine et al., 2018a*) was used to calculate the map to model FSC. The final model comprises residues 2–102, 154–231, 307–388, 410–476, and 489–697, for a total of 537 out of 724 residues in OSCA3.1 sequence. A segment of the BLD was modelled as poly-A helices and registry was tentatively assigned based on sequence and structure alignment to OSCA1.2 (*Jojoa-Cruz et al., 2018*): residues 234–246 and 280–303 assigned as IL2H2 and IL2H3, respectively. Pore profiles were predicted by CHAP (*Klesse et al., 2019*; *Rao et al., 2019*). Structure figures were made with PyMOL (*Schrödinger L, 2020*), UCSF Chimera (*Pettersen et al., 2004*) or UCSF ChimeraX (*Goddard et al., 2018*). Amino acid sequence alignment was obtained from Clustal omega (*Sievers et al., 2011*) and represented with ESPript3 (*Robert and Gouet, 2014*). Sequences used for the alignment were the

same as previous publication (*Jojoa-Cruz et al., 2018*), but the display was limited to the *Arabidopsis thaliana* OSCAs with solved structure.

## Generation of mutants, cell culture and transfections

A vector containing OSCA1.2 pIRES2-mCherry was used for electrophysiology experiments (*Murthy et al., 2018*). The W75K substitution was introduced using Q5 site-directed mutagenesis kit (New England Biolabs, NEB). Substitutions L80E, K435I, and K536I were introduced using QuikChange multi site-directed mutagenesis kit (Agilent). For the BLD chimera, OSCA1.2 pIRES2-mCherry vector and the BLD of OSCA3.1 were separately amplified by PCR using Q5 High-Fidelity 2 X Master Mix (NEB), followed by fragment assembly using Gibson Assembly Master Mix (NEB). Kits were used according to manufacturer's instructions.

Cell culture and transfection of HEK-P1KO cells for electrophysiology experiments were conducted as previously reported (*Jojoa-Cruz et al., 2018*).

## Electrophysiology

Patch-clamp experiments in transiently transfected HEK-P1KO cells were performed in standard whole-cell and cell-attached mode using a Multi-clamp700A amplifier (Axon Instruments) and followed the procedures described in our previous report (*Murthy et al., 2018*).

## Acknowledgements

We thank W Anderson for management the electron microscopy facility at Scripps Research, J Torres for help with data collection, and C Bowman, L Dong and JC Ducom for assistance with computation. We acknowledge members of the Ward laboratory. This work was supported by NIH grant R01 HL143297 and a Ray Thomas Edwards Foundation grant to ABW. AED was supported by grant R35 NS105067. Molecular graphics and analyses performed with UCSF Chimera and UCSF ChimeraX, developed by the Resource for Biocomputing, Visualization, and Informatics at the University of California, San Francisco, with support from National Institutes of Health R01-GM129325 and P41-GM103311, and the Office of Cyber Infrastructure and Computational Biology, National Institute of Allergy and Infectious Diseases.

## Additional information

### Funding

| Funder | Grant reference number | Author |
| --- | --- | --- |
| National Heart, Lung, and Blood Institute | R01 HL143297 | Andrew B Ward |
| National Institute of Neurological Disorders and Stroke | R35 NS105067 | Adrienne E Dubin |
| Ray Thomas Edwards Foundation | | Andrew B Ward |

The funders had no role in study design, data collection and interpretation, or the decision to submit the work for publication.

### Author contributions

Sebastian Jojoa-Cruz, Conceptualization, Formal analysis, Investigation, Writing - original draft, Writing – review and editing; Adrienne E Dubin, Conceptualization, Formal analysis, Investigation, Writing – review and editing; Wen-Hsin Lee, Formal analysis, Investigation, Writing – review and editing; Andrew B Ward, Conceptualization, Supervision, Funding acquisition, Writing – review and editing

### Author ORCIDs

Sebastian Jojoa-Cruz ⑩ http://orcid.org/0000-0002-4392-3898

Wen-Hsin Lee 🆔 http://orcid.org/0000-0001-9445-6671
Andrew B Ward 🆔 http://orcid.org/0000-0001-7153-3769

Reviewer #1 (Public review): https://doi.org/10.7554/eLife.93147.3.sa1
Reviewer #2 (Public review): https://doi.org/10.7554/eLife.93147.3.sa2
Reviewer #3 (Public review): https://doi.org/10.7554/eLife.93147.3.sa3
Author response https://doi.org/10.7554/eLife.93147.3.sa4

## Additional files

### Supplementary files
• MDAR checklist

### Data availability
Cryo-EM maps of OSCA3.1 in nanodiscs has been deposited to the Electron Microscopy Data Bank (EMDB) under accession number EMD-41911 with the LocalDeblur map as the primary map. The corresponding atomic coordinates have been deposited to the Protein Data Bank (PDB) under ID 8U53.

The following datasets were generated:

| Author(s) | Year | Dataset title | Dataset URL | Database and Identifier |
|---|---|---|---|---|
| Jojoa-Cruz S, Lee WH, Ward AB | 2023 | Cryo-EM map of OSCA3.1 in nanodiscs | https://www.ebi.ac.uk/emdb/EMD-41911 | Electron Microscopy Data Bank, EMD-41911 |
| Jojoa-Cruz S, Lee WH, Ward AB | 2023 | Atomic coordinates of OSCA3.1 in nanodiscs | https://www.rcsb.org/structure/8U53 | RCSB Protein Data Bank, 8U53 |

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
