## [Editor Report · eLife assessment]

The manuscript seeks to dissect the molecular underpinnings of poke and stretch activation in OSCA channels. While the structural and functional experiments are well done, and the authors present some **important** data, the reviewers identified weaknesses in experimental design and interpretation that render the data **incomplete** in supporting some of the main conclusions of the paper. Nevertheless, this work will be of interest to those working in the fields of mechanosensation, sensory biology, and ion channels.

---

## [Referee Report · Reviewer #1 (Public review)]

Summary:

The OSCA/TMEM63 channels have recently been identified as mechanosensitve channels. In a previous study, the authors found that OSCA subtypes (1, 2, and 3) respond differently to stretch and poke stimuli. For example, OSCA1.2 is activated by both poke and stretch, while OSCA3.1, responds strongly to stretch but poorly to poke stimuli. In this study the authors use cryo-EM, mutagenesis, and electrophysiology to dissect the mechanistic determinants that underlie the channels' ability to respond to poke and stretch stimuli.

The starting hypothesis of the study is that the mechanical activation of OSCA channels relies on the interactions between the protein and the lipid bilayer and that the differential responses to poke and stretch might stem from variations in the lipid-interacting regions of OSCA proteins. The authors specifically identify the amphipathic helix (AH), the fenestration, and the Beam Like Domain (BLD) as elements that might play a role in mechanosensing.

The authors use solid methodology to show that poke and stretch responses likely use different mechanisms in OSCA channels and that the poke response can be uncoupled from the stretch response in OSCA1.2 by mutations in the AH and the positively charged residues in the fenestration. However, the study falls short of explaining why OSCA3.1 does not respond efficiently to poke stimuli. This question is particularly important as the AH residues that are important for the poke response in OSCA1.2 are present in OSCA3.1.

Unfortuntately, due to staffing issues, the authors were unable to perform additional experiments that would address some of the critical issues that were brought up during peer review. Nevertheless, the structural and functional data presented is of high quality and the findings on OSCA1.2 will be of interest to anyone working in the fields of mechanosensation, sensory biology, and ion channels.

---

## [Referee Report · Reviewer #2 (Public review)]

Jojoa-Cruz et al. have submitted a revised manuscript and their responses to reviewers' comments on the major weaknesses of the paper and recommendations. The authors have made minimal changes to the manuscript itself, which highly resembles the initial submission. Most concerningly, the authors appeared to agree with reviewers' comments, but did not and are not going to carry out any of the recommended experiments, including electrophysiology [Reviewer 2- major point 3, recommended point 5; Reviewer 3- recommended point 4] and western blot [Reviewer 3- recommended point 3], by explaining that they have left the lab. The major weakness and issues raised in the previous review process therefore remain in the current version of the manuscript.

Moreover, in the public review major weakness, the reviewer pointed out issues on the inadequacy of the functional validation on the structural domains based on mutagenesis of OSCA1.2 vs. OSCA3.1 and using poke and stretch assays, as well as weakness in the corresponding mechanistic interpretation of the functional data. These issues need to be addressed or improved to a certain extent through revised study design and execution of experiments.

---

## [Referee Report · Reviewer #3 (Public review)]

Summary:

Jojoa-Cruz et al provide a new structure of At-OSCA3.1. The structure of OSCA 3.1 is similar to previous OSCA cryo-em structures of both OSCA3.1 and other homologues validating the new structure. Using the novel structure of OSCA3.1 as a guide they created several point mutations to investigate two different mechanosensitive modalities: poking and stretching. To investigate the ability of OSCA channels to gate in response to poking they created point mutations in OSCA1.2 to reduce sensitivity to poking based on the differences between the OSCA1.2 and 3.1 structures. Their results suggest that two separate regions are responsible for gating in response to poking and stretching.

Strengths:

Through a detailed structure based analysis, the authors identified structural differences between OSCA3.1 and OSCA1.2. The use of technically sound data supports the hypothesis that poking and stretching are sensed by two unique regions in the protein. These subtle structural changes between homologues identify regions in the amphipathic helix and near the pore that are essential for gating of OSCA1.2 in response to poking and stretching. Mutations in the AH of OSCA1.2 decrease the sensitivity to poking stimulus however these mutations have similar stretch activated currents to the WT. The point mutations described in the manuscript will set the foundation for investigations into how these two channels sense tension using different regions of structurally similar proteins.

Weaknesses:

Mutations in the amphipathic helix at W75 and L80 show reduced gating in response to poking stimuli. The gating observed occurs at poking depths similar to cellular rupture, the similarity in depths suggests that these mutations could be a complete loss of functions.

---

## [Author Response]

The following is the authors’ response to the original reviews.

**Public Reviews:**

**Reviewer #1 (Public Review):**
Summary:The OSCA/TMEM63 channels have recently been identified as mechanosensitive channels. In a previous study, the authors found that OSCA subtypes (1, 2, and 3) respond differently to stretch and poke stimuli. For example, OSCA1.2 is activated by both poke and stretch, while OSCA3.1, responds strongly to stretch but poorly to poke stimuli. In this study, the authors use cryo-EM, mutagenesis, and electrophysiology to dissect the mechanistic determinants that underlie the channels' ability to respond to poke and stretch stimuli.The starting hypothesis of the study is that the mechanical activation of OSCA channels relies on the interactions between the protein and the lipid bilayer and that the differential responses to poke and stretch might stem from variations in the lipid-interacting regions of OSCA proteins. The authors specifically identify the amphipathic helix (AH), the fenestration, and the Beam Like Domain (BLD) as elements that might play a role in mechanosensing.The strength of this paper lies in the technically sound data - the structural work and electrophysiology are both very well done. For example, the authors produce a high-resolution OSCA3.1 structure which will be a useful tool for many future studies. Also, the study identifies several interesting mutants that seemingly uncouple the OSCA1.2 poke and stretch responses. These might be valuable in future studies of OSCA mechanosensation.

However, the experimental approach employed by the authors to dissect the molecular mechanisms of poke and stretch falls short of enabling meaningful mechanistic conclusions. For example, we are left with several unanswered questions surrounding the role of AH and the fenestration lipids in mechanosensation: Is the AH really important for the poke response if mutating residues conserved between OSCA1.2 and OSCA3.1 disrupts the OSCA1.2 ability to respond to poke but mutating the OSCA1.2 AH to resemble that of OSCA3.1 results in no change to its "pokability"? Similar questions arise in response to the study of the fenestrationlining residues.

We thank the reviewer for their feedback. We believe that the different OSCA1.2 mutants on their own suggest an involvement of the AH and fenestration-lining residues in its mechanosensitive response. We attribute the inability to restore the poke response of OSCA3.1 with similar mutations to its inherent high threshold to this particular stimulus and perhaps other structural differences, or a combination of them, that we did not probe in this study. We agree more work is required in the field to address these remaining questions and further dissect the difference between poke and stretch responses.

**Reviewer #2 (Public Review):**
Summary:Jojoa-Cruz et al. determined a high-resolution cryo-EM structure in the *Arabidopsis thaliana* (At) OSCA3.1 channel. Based on a structural comparison between OSCA3.1 and OSCA1.2 and the difference between these two paralogs in their mechanosensitivity to poking and membrane stretch, the authors performed structural-guided mutagenesis and tested the roles of three structural domains, including an amphipathic helix, a beam-like domain, and a lipid fenestration site at the pore domain, for mechanosensation of OSCA channels.Strengths:The authors successfully determined a structure of the AtOSCA3.1 channel reconstituted in lipid nanodiscs by cryo-EM to a high resolution of 2.6 Å. The high-resolution EM map enabled the authors to observe putative lipid EM densities at various sites where lipid molecules are associated with the channel. Overall, the structural data provides the information for comparison with other OSCA paralogs.In addition, the authors identified OSCA1.2 mutants that exhibit differential responses to mechanical stimulation by poking and membrane stretch (i.e., impaired response to poke assay but intact response to membrane stretch). This interesting behavior will be useful for further study on differentiating the mechanisms of OSCA activation by distinct mechanical stimuli.Major weakness:The major weaknesses of this study are the mutagenesis design and the functional characterization of the three structural domains - an amphipathic helix (AH), a beam-like domain (BLD), and the fenestration site at the pore, in OSCA mechanosensation.(1) First of all, it is confusing to the reviewer, whether the authors set out to test these structural domains as a direct sensor(s) of mechanical stimuli or as a coupling domain(s) for downstream channel opening and closing (gating). The data interpretations are vague in this regard as the authors tend to interpret the effects of mutations on the channel 'sensitivity' to different mechanical stimuli (poking or membrane stretch). The authors ought to dissect the molecular bases of sensing mechanical force and opening/closing (gating) the channel pore domain for the structural elements that they want to study.

We agree with the reviewer that our data are unable to distinguish the transduction of a mechanical stimulus and channel gating. We set up to determine whether these features were involved in the mechanosensitive response. However, as the reviewer points out, evaluating whether they work as direct sensors or coupling domains would require a more involved experimental design that lies beyond the scope of this work. Thus, we do not claim in our study whether these features act as direct sensors of mechanosensitive stimuli or as coupling domains, only their involvement.

Furthermore, the authors relied on the functional discrepancies between OSCA1.2 (sensitive to both membrane poking and stretch) and OSCA3.1 (little or weak sensitivity to poking but sensitive to membrane stretch). But the experimental data presented in the study are not clear to address the mechanisms of channel activation by poking vs. by stretch, and why the channels behave differently.

We had hoped that when we switched regions of the OSCA1.2 and OSCA3.1 channels we would abolish poke-induced responses in OSCA1.2 and confer poke-induced sensitivity to OSCA3.1. We agree with the reviewer that we were not able to pinpoint the reason or multiple reasons, as it could be a compounded effect of several differences, that caused OSCA3.1 higher threshold and thus we could not confer to it an OSCA1.2-like phenotype. Yet, we shed some light on some of the structural differences that appear to contribute to OSCA3.1 behavior, as mutagenesis of OSCA1.2 to resemble this channel led to OSCA3.1-like phenotype.

(2) The reviewer questions if the "apparent threshold" of poke-induced membrane displacement and the threshold of membrane stretch are good measures of the change in the channel sensitivity to the different mechanical stimuli.

The best way to determine an accurate measure of sensitivity to mechanical stimuli is stretch applied to a patch of membrane. There are more complicating factors that influence the determination of "apparent threshold" in the whole cell poking assay, including visualizing when the probe first hits the cell (very difficult to see). With that said, the stretch assay has its own issues such as the creep of the membrane into the pipette glass which we try to minimize with positive pressure between tests.

(3) Overall, the mutagenesis design in the various structural domains lacks logical coherence and the interpretation of the functional data is not sufficient to support the authors' hypothesis. Essentially the authors mutated several residues on the hotspot domains, observed some effects on the channel response to poking and membrane stretch, then interpreted the mutated residues/regions are critical for OSCA mechanosensation. Examples are as follows.In the section "Mutation of key residues in the amphipathic helix", the authors mutated W75 and L80, which are located on the N- and C-terminal of the AH in OSCA1.2, and mutated Pro in the OSCA1.2 AH to Arg at the equivalent position in OSCA3.1 AH. W75 and L80 are conserved between OSCA 1.2 and OSCA3.1. Mutations of W75 and/or L80 impaired OSCA1.2 activation by poking, but not by membrane stretch. In comparison, the wildtype OSCA3.1 which contains W and L at the equivalent position of its AH exhibits little or weak response to poking. The loss of response to poking in the OSCA1.2 W/L mutants does not indicate their roles in pokinginduced activation.Besides, the P2R mutation on OSCA1.2 AH showed no effect on the channel activation by poking, suggesting Arg in OSCA3.1 AH is not responsible for its weak response to poking. Together the mutagenesis of W75, L80, and P2R on OSCA1.2 AH does not support the hypothesis of the role of AH involved in OSCA mechanosensation.

Mutagenesis of OSCA1.2 in the amphipathic helix for residues W75 and L80 suggests a role of the helix in the poke response in OSCA1.2, regardless of OSCA3.1 having the same residues. Furthermore, the lack of alteration in the response for mutant P77R suggests that specific residues of the helix are involved in this response and is not a case where any mutation in the helix will lead to a loss of function.

OSCA3.1 WT exhibits a high-threshold response (near membrane rupture) in the poke assay without any mutations, and this could be due to other features, for example, the residues lining the membrane fenestration, as well as features not identified/probed in this study. We agree with the reviewer that the differences in the AH do not explain the different response to poke in OSCA1.2 and OSCA3.1, and we have added this statement explicitly in the discussion for clarification (line #251-252).

In the section "Replacing the OSCA3.1 BLD in OSCA1.2", the authors replaced the BLD in OSCA 1.2 with that from OSCA3.1, and only observed slightly stronger displacement by poking stimuli. The authors still suggest that BLD "appears to play a role" in the channel sensitivity to poke despite the evidence not being strong.

We agree with the reviewer that the experiments carried out show little difference between the response of OSCA1.2 WT and OSCA1.2 with OSCA3.1 BLD, and we have stated so (line #259: “Substituting the BLD of OSCA1.2 for that of OSCA3.1 had little effect on poke- or stretchactivated responses. Although these results suggest that the BLD may not be involved in modulating the MA response of OSCA1.2…”). However, the section of the discussion that the reviewer points out also considers evidence provided by recent reports from Zheng, et al. (Neuron, 2023) and Jojoa-Cruz, et al. (Structure, 2024) and we suggest an hypothesis to reconcile our findings with these new evidence.

OSCA1.2 has four Lys residues in TM4 and TM6b at the pore fenestration site, which were shown to interact with the lipid phosphate head group, whereas two of the equivalent residues in OSCA3.1 are Ile. In the section "Substitution of potential lipid-interacting lysine residues", the authors made K435I/K536I double mutant for OSCA1.2 to mimic OSCA3.1 and observed poor response to poking but an intact response to stretch. Did the authors mutate the Ile residues in OSCA3.1 to Lys, and did the mutation confer channel sensitivity to poking stimuli resembling OSCA1.2? The reviewer thinks it is necessary to perform such an experiment, to thoroughly suggest the importance of the four Lys residues in lipid interaction for channel mechanoactivation.

We thank the reviewer for this suggestion. We agree that the suggested experiments will further improve the quality of the results, but we are no longer able to perform such experiments.

**Reviewer #3 (Public Review):**
Summary:Jojoa-Cruz et al provide a new structure of At-OSCA3.1. The structure of OSCA 3.1 is similar to previous OSCA cryo-em structures of both OSCA3.1 and other homologues validating the new structure. Using the novel structure of OSCA3.1 as a guide they created several point mutations to investigate two different mechanosensitive modalities: poking and stretching. To investigate the ability of OSCA channels to gate in response to poking they created point mutations in OSCA1.2 to reduce sensitivity to poking based on the differences between the OSCA1.2 and 3.1 structures. Their results suggest that two separate regions are responsible for gating in response to poking and stretching.Strengths:Through a detailed structure-based analysis, the authors identified structural differences between OSCA3.1 and OSCA1.2. These subtle structural changes identify regions in the amphipathic helix and near the pore that are essential for the gating of OSCA1.2 in response to poking and stretching. The use of point mutations to understand how these regions are involved in mechanosensation clearly shows the role of these residues in mechanosensation.Weaknesses:In general, the point mutations selected all show significant alterations to the inherent mechanosensitive regions. This often suggests that any mutation would disrupt the function of the region, additional mutations that are similar in function to the WT channel would support the claims in the manuscript. Mutations in the amphipathic helix at W75 and L80 show reduced gating in response to poking stimuli. The gating observed occurs at poking depths similar to cellular rupture, the similarity in depths suggests that these mutations could be a complete loss of function. For example, a mutation to L80I or L80Q would show that the addition of the negative charge is responsible for this disruption not just a change in the steric space of the residue in an essential region.

We thank the reviewer for this suggestion. We agree that the suggested experiments will further improve the quality of the results, but we are unable to perform such experiments due to the authors having moved on from the respective labs.

**Recommendations for the authors:**

**Reviewer #1 (Recommendations For The Authors):**
I have several questions regarding some of the aspects of your study:Mutation of the hydrophobic W75 and L80 in OSCA1.2 to charged residues significantly decreases the poke response in OSCA1.2 without affecting the stretch response. However, W75 and L80 are also present in OSCA3.1, which does not respond efficiently to poke. You conclude that these two residues are important for the poke response, but do not delve into why, if these residues are important, OSCA3.1 is not poke-sensitive.In addition, mutation of the OSCA1.2 AH to resemble that of OSCA3.1 does not produce channels that are less poke-sensitive. Given the data presented, if AH were a universal "poke sensor", one could also expect WT OSCA3.1 to exhibit a robust poke response, like OSCA1.2. Here I think it would be important to explain in more detail how this data might fit together.

We thank the reviewer for bringing up this issue. We decided to test the importance of the AH due to the presence of similar structures in other mechanosensitive channels. Our data showed that single and double mutants of the AH of OSCA1.2 affected its poke response but not stretch. This supports the idea of the AH involvement in the poke response. Yet, we agree that the differences in the AH between OSCA1.2 and OSCA3.1 (P77R mutation) do not explain the higher threshold of OSCA3.1, we have explicitly added this in line #255. The particular OSCA3.1 phenotype may be due to other differences in the structure, for example, differences in the membrane fenestration area, or a combined effect of several differences, which we believe is more likely.

I also have some questions about the protein-lipid interactions in the fenestration. A lipid has been observed in this location in both OSCA1.2 and OSCA3.1 structures. Mutation of the two OSCA1.2 lysines to isoleucines results in channels that are resistant to poke which leads to the conclusion that the interactions between the fenestration lysines and lipids are important for the poke response.Here, there are several questions that arise but are not answered:It is not shown what happens when OSCA3.1 isoleucines are mutated to lysines - do these mutants result in poke-able channels? Is the OSCA3.1 mechanosensing altered?

We performed a preliminary test on OSCA3.1 I423K/I525K double mutant (n = 3). However, we did not see an increase in poke sensitivity. We attributed this to other unexplored differences in OSCA3.1 having an effect in channel mechanosensitivity.

It is implied that the poke response is predicated on the lysine-lipid interaction. However, lipid densities are present in both OSCA1.2 and OSCA3.1 structures, indicating that both fenestrations interact with lipids. How can we be certain that the mutation of lysine to isoleucine does not disrupt an inter-protein interaction rather than a protein-lipid one? For example, the K435I mutation might disrupt interactions with D523 or the backbone of G527?

The reviewer brings up a good point. We believe the phenotype seen is due to a different strength in the interaction between lipids and proteins, however, disrupted interaction with other residues is a valid alternative explanation. We agree that the suggested experiments will further clarify the results, but we are unable to perform such experiments due to the authors having moved on from the respective labs.

Similarly, the effects of single lysine-to-isoleucine (K435I or K536I) mutations are not explored.The observed effect might be caused by only one of these substitutions.

We thank the reviewer for this suggestion. We agree that the suggested experiments will further improve the quality of the results, but we are unable to perform such experiments due to the authors having moved on from the respective labs.

I also wanted to take this opportunity to ask a couple of philosophical (?) questions about using a mammalian system to study ion channels that have evolved to function in plants. Your study highlights the intimate relationship between the lipid bilayer and protein function/mechanosensitivity. Plant cells contain high levels of sterols and cerebrosides that would significantly affect both cell stiffness and the specific interactions that can be formed between the protein and the lipid bilayer. I wonder if the properties of the lipid bilayer might shift the thresholds for poke and/or stretch stimuli and if structural elements that do not appear to have a major role in mechanosensation in a mammalian cell (e.g., BLD) might be very influential in a lipid environment that more closely resembles that of a plant?Conversely, is it possible that OSCA channels are not poke-sensitive in plant cells?These questions are beyond the scope of your study, but they might be a nice addition to your discussion.

The reviewer poses a great question. Electrophysiological approaches for studying plant mechanosensitive channels suffer the limitation of not being able to fully reconstitute the environment of a plant cell. To be able to patch the cell, the cell wall needs to be disposed of, which eliminates the tension generated from this structure onto the membrane. In that sense, performing these assays in plant cells or another system would not give us a fully accurate picture of the physiological thresholds of these channels. Given this limitation, we performed our study with mammalian cells given our expertise with them. Like the reviewer, we are also intrigued by the effect of different membrane compositions on the behavior of OSCA channels and how these channels will behave under physiological conditions, but we agree with the reviewer that these questions are out of the scope of our work. To address this point, in line #294 we have added: “It is also important to note that the membrane of a plant cell contains a different lipid composition than that of HEK293 cells used in our assays, and thus these lipids, or the plant cell wall, may alter how these channels respond to physiological stimuli.”

Line 313 For structural studies, human codon-optimized OSCA3.1. Could you please clarify what this means?

We have changed the phrase to “For structural studies, the OSCA3.1 (UniProt ID: Q9C8G5) coding sequence was synthesized using optimized codons for expression in human cells and subsequently cloned into the pcDNA3.1 vector” in line #327 to clarify this sentence.

As a final comment, in the methods you use references to previously published work. I would strongly encourage you to replace these with experimental details.

We understand the reviewer’s argument. However, this article falls under eLIFE’s Research Advances and will be linked to the original published work to which we reference the method. As suggested in the guidelines for this type of article, we only described the methods that were different from the original paper.

**Reviewer #2 (Recommendations For The Authors):**
(1) In line 85, provide C-alpha r.m.s.d. values for the structural alignment among OSCA3.1, OSCA1.1, and OSCA1.2 protomers.

As requested, we have added the C-alpha RMSD in line #86.

(2) In line 90, should the figure reference to Fig. 1d be Fig. 1e?

We thank the reviewer for catching this error. We have corrected it in the manuscript.

(3) In lines 89-94, what putative lipid is it resolved in the OSCA3.1 pore? Can the authors assign the lipid identity? Is this the same or different from the lipids resolved in OSCA1.2, OSCA1.1, and TMEM63?

In the model, we have built the lipid as palmitic acid to represent a lipid tail, but the resolution in this area makes it difficult to ascertain the identity of said lipid, hence we cannot compare to lipids in other orthologs.

(4) In lines 115-121, the authors describe the presence of AHs and their functional roles in MscL and TMEM16. It will be more informative if the authors can add figures to show the structure of MscL and highlight the analogous AH. In addition, the current Supplementary Fig. 6 is not informative so it should be improved. It is not clear to the reviewer why that stretch of helix in TMEM16 is equivalent or analogous to the AH in OSCAs, either sequence alignment or a detailed structural alignment is helpful to address this point. Also, in lines 120-121, it says this helix in TMEM16 "does not present amphipathic properties", please show the sequence or amphipathicity of the helix.

We thank the reviewer for the feedback on this figure. Supplementary Fig. 6 has been thoroughly modified to address the reviewer’s concerns. We now include a panel showing the structure of MscL and its amphipathic helix. We have modified the alignment of OSCA3.1 to a TMEM16 homolog to make clearer the homologous positioning of the helices in question and zoom in to show their sequences.

(5) In discussion, lines 249-257, the authors referred to a recent study that suggested three evolutionarily coupled residue pairs located on BLD and TM6b. The authors speculate that the reason they did not observe a significant effect of channel response to poke/stretch stimuli in the BLD swapping between OSCA1.2 and 3.1 is due to the 2 of 3 salt bridges remaining for the residue pairs. To test the importance of these residue pairs and their coupling for channel gating, instead of swapping the entire BLD, can the authors systematically mutate the residue pairs, disrupt the salt-bridge interactions, and analyze the effect on channel response to mechanical force?

We thank the reviewer for this suggestion. We agree that the suggested experiments will further improve the quality of the results, but we are unable to perform such experiments due to the authors having moved on from the respective labs.

(6) The reviewer suggests the authors tone down the elaboration of polymodal activation of OSCA by membrane poking and stretch.

We believe the idea of polymodal activation is sufficiently toned down as we only postulate it as a possibility and following we give an alternative explanation based on methodological limitations: “Nonetheless, the discrepancy could be due to inherent methodological differences between these two assays, as whole-cell recordings during poking involve channels in inaccessible membranes (at the cell-substrate interface) and channel interactions with extracellular and intracellular components, while the stretch assay is limited to recording channels inside the patch.”

(7) In lines 81-83, the authors described the BLD as showing increased flexibility, and the EM map at this region is less well resolved for registry assignment. In the method for cryo-EM image processing and Supplementary Fig. 1, the authors only carried out 3D refinement and classification at the full channel level. Have the authors attempted to do focus refinement or classification at the BLD domain in order to improve the local resolution or to sort out conformational heterogeneity? The reviewer suggests doing so because the BLD domain is a hot spot that the authors have proposed to play an important role in OSCA mechanosensation. Conformational changes identified in this region might provide insights into its role in the channel function.

We thank the reviewer for this suggestion. We have performed focused classification on the BLD with and without surrounding regions and, in our hands, it did not improve the resolution or provide further insights.

**Reviewer #3 (Recommendations For The Authors):**
Here are a few specific minor corrections that should be addressed(1) In lines 117-135, in the discussion of Figure 2, the data shows an apparent increase in the poking threshold to gate W75K/L80E. The substantial increase in the depth required to gate the channel suggests that these channels are less sensitive to poking. Would it be possible to compare the depth at which these two patches show activity and the depth at which the other 22 cells ruptured? Line 161 mentions that the rupture threshold of HEK cells is close to the gating of OSCA3.1 at 13.8 µm.

The distance just before the cell ruptured in 22 cells with no response was 12.5 +/- 2.5 um. The distance at which the cells ruptured was 0.5 um more (13 +/- 2.5 n=22). We have added this last value in line #137.

(2) Would it be possible in Figures 2 panels b and c, 3, and figure 4 to label the WT as WT OSCA1.2?

We thank the reviewer for pointing this out. We agree this modification will improve the clarity of the figures and have changed the figures to follow the reviewer’s suggestion.

(3) Can you provide a western blot of the mutations described in Figure 2? This would provide insight into the amount of protein at the cell surface and available to respond to poking, the stretch data shows that these channels are in the membrane but does not show if they are in the membrane in similar quantities.

We thank the reviewer for this suggestion. We agree that the suggested experiments will further improve the quality of the results, but we are unable to perform such experiments due to the authors having moved on from the respective labs.

(4) The functional differences between the two channels are projected to be tied to several distinct point mutations, however, the data could be strengthened by additional point mutations at all sites to show that the phenotypes are due to the mutations specifically not just any mutation in the region.

We thank the reviewer for this suggestion. We agree that the suggested experiments will further improve the quality of the results, but we are unable to perform such experiments due to the authors having moved on from the respective labs.